# Potential Additives in Natural Rubber-Modified Bitumen: A Review

**DOI:** 10.3390/polym15081951

**Published:** 2023-04-20

**Authors:** Nurul Farhana Rohayzi, Herda Yati Binti Katman, Mohd Rasdan Ibrahim, Shuhairy Norhisham, Noorhazlinda Abd Rahman

**Affiliations:** 1Department of Civil Engineering, College of Engineering, Universiti Tenaga Nasional, Putrajaya Campus, Jalan Ikram-Uniten, Kajang 43000, Malaysia; 2Institute of Energy Infrastructure, Universiti Tenaga Nasional, Putrajaya Campus, Jalan Ikram-Uniten, Kajang 43000, Malaysia; 3Centre for Transportation Research, Department of Civil Engineering, Engineering Faculty, Universiti Malaya, Kuala Lumpur 50603, Malaysia; 4School of Civil Engineering, Engineering Campus, Universiti Sains Malaysia, Penang 14300, Malaysia

**Keywords:** energy, sustainability, natural rubber, bitumen, additives, rubberised bitumen, rubber modified bitumen

## Abstract

Conventional bitumen pavement is no longer suitable for handling increasing loads and weather variations, which cause road deterioration, Thus, the modification of bitumen has been suggested to counter this issue. This study provides a detailed assessment of various additives for modifying natural rubber-modified bitumen used in road construction. This work will focus on the use of additives with cup lump natural rubber (CLNR), which has recently started to gain attention among researchers, especially in rubber-producing countries such as Malaysia, Thailand and Indonesia. Furthermore, this paper aims to briefly review how the addition of additives or modifiers helps elevate the performance of bitumen by highlighting the significant properties of modified bitumen after the addition of modifiers. Moreover, the amount and method of application of each additive are discussed further to obtain the optimum value for future implementation. On the basis of past studies, this paper will review the utilisation of several types of additives, including polyphosphoric acid, Evotherm, mangosteen powder, trimethyl-quinoline and sulphur, and the application of xylene and toluene to ensure the homogeneity of the rubberised bitumen. Numerous studies were conducted to verify the performance of various types and compositions of additives, particularly in terms of physical and rheological properties. In general, additives enhance the properties of conventional bitumen. Future research should investigate CLNR because studies on its utilisation are limited.

## 1. Introduction

One of the concerns of the transportation system is to deliver the optimal service level through effective road maintenance with utmost safety at the lowest cost. The road network is one of the most significant modes of transportation that enables people to travel from one location to another. However, weather significantly affects the performance of pavement. Hence, understanding how pavement performs under various weather conditions and load situations is vital. With long-term exposure to sunlight, oxygen, rain and traffic loads on bitumen pavement, the bitumen mixture becomes brittle and hard, causing fractures and other types of damage, which reduce pavement serviceability.

Aside from loading and weathering conditions, the use of materials with inadequate design or poor construction methods can contribute to several pavement distresses. Some common distresses include pavement cracks, patching and potholes, surface deformation and surface defect, moisture damage and ravelling. The rheological characteristics of the bitumen materials used in pavement construction are connected to primary pavement distresses, such as rutting at high temperatures, thermal cracking at low temperatures and fatigue cracking due to recurrent traffic loads. Conventional pavement with unmodified bitumen that follows standard design procedures faces challenges in maintaining its performance not only during peak traffic conditions but also during severe environmental conditions. Thus, researchers, engineers and road authorities need to look for reliable and cost-effective mitigation methods that could improve the engineering properties of conventional bitumen and lengthen the life span of bitumen pavements [1]. 

Paving technologists have tried to enhance bitumen pavements by using innovative methods to design and develop durable pavements. One such effort is bitumen modification, which aims to transform and improve the characteristics of bitumen to optimise the long-term performance of pavements and minimise temperature dependence, oxidative bitumen hardening and the moisture susceptibility of bitumen mixes [2]. Previous studies obtained encouraging results for polymer-modified bitumen, highlighting the application of polymers in bitumen, which has sparked much interest in recent years. The fundamental benefit of polymer technology is that it improves the adhesive qualities between the bitumen and the aggregate, thus significantly increasing the performance of conventional bitumen [3]. The type of modifier and bitumen generally affects the properties of the modified bitumen [1]. Polymer-modified bitumen has demonstrated positive results in the lab and in the field, and work is ongoing to establish a link between laboratory test findings and field performance. Shaffie et al. [3] mentioned that polymer-modified bitumen improved aggregate surface roughness, adhesion and degree of cohesion by forming an aggregate coating material, resulting in superior bitumen mixes.

This review paper aims to evaluate in detail the potential additives used in natural rubber-modified bitumen (NRMB) to enhance the physical, chemical and rheological properties of the mixtures for better road performance. Some additives are discussed, such as polyphosphoric acid (PPA), Evotherm, mangosteen powder (MPP), trimethyl-quinoline (TMQ) and sulphur, which are some of the most common additives used in rubber modifiers, as indicated in previous studies. We outline how they are implemented in bitumen modification and their significant contributions to the enhancement of modified bitumen. Moreover, to ensure the homogeneity of the rubber mixture, this paper focuses on the inclusion of xylene and toluene as a solvent for rubber treatment. The details on its application and its impact on the rubber mixtures are discussed to give a brief view for future research on the importance of rubber treatment and how it affects the performance of modified bitumen. 

### Natural Rubber as a Bitumen Modifier

Road construction is a key strategy for rubber-producing countries to increase utilisation and stabilise the price of natural rubber (NR). NR is a renewable resource with a high molecular weight of hydrocarbon polymers, and is an environmentally friendly elastomer that has not only excellent elasticity but also good cyclical load performance [2,4]. Its complex molecular chains create strong bonds and structures, producing a number of unique traits, such as elasticity, the capacity to withstand high tension, thermal stability and resiliency, which contribute to its use in polymer-modified bitumen [2]. In South Asia, NR is produced by the native tree *Hevea brasiliensis*, which, when combined with other materials that act as insulating membranes and additives, is has been widely utilised in various products in numerous sectors such as the tire industry, footwear and construction [5]. 

Rubber prices have steadily declined in recent years because of the global commodities crisis. The fluctuation in the price of NR has resulted in an excess of rubber production. Thus, its application should be varied to sustain the rubber sector. NR is broadly utilised because of its high stretch ratio, excellent ductile response, elasticity, ability to sustain external loads, resistance to permanent deformation and superior water resistance. Its strong durability and high stretch ratio make it generally elastic in terms of its physical traits. Its viscoelastic properties, such as that exhibited by polyisoprene, is due to complex molecular chains that form nearly linear chains when loaded and return to their original positions when the load is released. Thus, it is commonly used even though it has both liquid and solid properties [6,7]. Even though its features contribute to improving water resistance, the use of NR is consistent with the environmental requirement to use recycled materials and environmentally friendly alternatives to conserve finite natural resources [5].

Natural resource depletion is a global issues that causes concern among governments and municipal authorities. Promoting recycling and seeking sustainable building materials are solutions to this crisis. The use of polymer-modified bitumen is required to minimise pavement distresses and improve the viscoelastic characteristics of bitumen. NR is an effective polymer for modifying bitumen. It offers better fatigue resistance than most elastomers and less heat build-up. It is also more economical, simple to make, and has a broader working temperature range and minimal creep. NR is a renewable resource that contributes to improving shear resistance, grips bitumen firmly and opposes the flow of bitumen. It also aids in the dissipation of produced stresses. Its inherent elastomer characteristics may contribute to improving long-term pavement performance characteristics such as resistance to rutting, fatigue cracking and stripping. When NR is placed in hot bitumen, the principle of rubber–bitumen interaction indicates that rubber particles absorb the substances with identical solubility values and swell quickly. Rubber particles can dissolve in liquid bitumen because of weak cross-links between the elastomer chains, and are therefore commonly used to modify bitumen [8].

The use of NR also contributes to the sustainability aim of the Sustainable Development Goals (SDGs). Some of the SDGs related to the application of NR in road construction are SDG 9: Industry, Innovation and Infrastructure, SDG 11: Sustainable Cities and Communities and SDG 12: Responsible Consumption and Production. The implementation of NR aims to provide sustainable industrialisation specifically in pavement construction, which benefits small-scale industries (rubber smallholders) and enterprises for road construction works. Moreover, NR is a safe and affordable modifier of bitumen for sustainable urbanisation with lower emission to achieve a clean environment. This method is expected to fulfil a longer road framework programme for sustainable development and management that benefits both economic growth and the public. 

Ansari et al. [9] stated that NR has several advantages for roadway pavement construction. As a cheap and environmentally friendly biopolymer, it helps reduce the reliance on petroleum-based modifiers. As a result, widespread usage of NR may boost industrial economic growth and extend the life span of roads with minimal maintenance costs, which would greatly increase ride quality. Furthermore, rubberised mixtures can withstand several load cycles without breaking at a lower strain [10]. The crystallised nature of bitumen, which has rigid traits and cannot be stretched, promotes formation of cracks at a higher strain. The performance of rubberised mixtures using NR is better than that of the conventional mixture. NR can be used as a modifier to improve the performance of the pavement and promote local rubber production, particularly in countries with large NR production, such as Thailand, Malaysia and Indonesia. Primarily, referring to Asvitha Valli and Kolathayar [11], NR absorbs vibrations well. Moreover, it is insoluble in water, acetone, dilute acids and alkalis, and it has high tensile and tear strength with great fatigue resistance.

NR has two main types: natural rubber latex (NRL) and a new type of NR known as cup lump natural rubber (CLNR) (Figure 1), which is produced when latex is allowed to coagulate under bacterial attack [12,13]. NRL polymer can be used as a modifier to enhance bitumen’s physical qualities, such as increasing the softening point, reducing penetration while enhancing the penetration index, and improving the properties of short- and long-term ageing [3,14,15]. NRL also addresses the common problems of bitumen, such as fatigue, permanent deformation, fracture strength and thermal cracking resistance, helping improve the durability of bitumen pavement [5,16]. Moreover, the addition of NRL improves the rheological properties by improving the rutting factor (G^∗^/sin δ), enabling bitumen to resist deformation on the road surface despite high traffic loading, thus achieving better rutting and fatigue resistance [16,17]. Latex-modified bitumen does not experience phase separation during hot-temperature storage due to improved viscoelastic capabilities. Morphological analysis revealed that the material has a homogeneous dispersed network [14,18]. 

The use of CLNR in road construction is anticipated to increase domestic rubber demand because of its cheap manufacturing costs and reduced water consumption and labour cost. It can be created either by evaporating water from fresh latex in a collecting cup or by coagulating latex with acid. Collecting cups on rubber trees are used to gather the cup lumps daily. This raw material is primarily sorted depending on its moisture content to produce different grades of rubber. This review paper aims to look at the implementation of CLNR in more detail. The next subtopics will focus on the application of CLNR. 

## 2. Cup Lump Natural Rubber

Cup lump is obtained when fresh latex is extracted by tapping into a long cut made in a rubber tree and letting the drips pour into a plastic cup. The substance is then mixed with formic acid to coagulate the latex. The process continues for several days until the required amount is reached [19]. Physically, CLNR consists of water, rubber and non-rubber material. The moisture content of CLNR often varies locally because of surface evaporation, which creates a moisture gradient that encourages diffusion toward the drier surface. According to Pamornnak et al. [19], the volume of CLNR generally ranges between 500 and 1500 cm^3^ depending on the cup used for collection. Normally, CLNR is around 76 mm high, with a diameter of 38 mm at the bottom and 101 mm at the top. It has a low water content and chemical properties identical to NRL, which have prompted researchers to explore its use in bitumen [20]. 

S. Abdulrahman et al. [4] mentioned that CLNR content significantly affects the storage stability and viscosity of the cup-lump-modified bitumen (CLMB). The rise in bitumen stiffness is proportional to the increase in mixing and compaction temperatures. In addition, the use of CLNR with additives produces excellent rutting resistance, improved rheological properties, better viscoelastic response at higher temperature, improved tensile property, moisture resistance and enhanced conventional properties of the base bitumen, such as penetration, softening point and temperature susceptibility [21,22,23,24]. An analysis of physical and rheological properties showed that the presence of elastic NR within the bitumen network increases the softening, viscosity and stiffness, and it reduces penetration and temperature susceptibility, consequently enhancing the rutting property [22,23,24]. Othman et al. [25] conducted an on-site evaluation and found that cup-lump-modified asphalt (CMA) pavement performed better in surface conditions with a high roughness value, structural condition, skid resistance and dynamic creep. 

The physical properties of the rubber have a significant influence on the composition and viscosity of bitumen; as a result, the physicomechanical properties of bitumen pavement are influenced by the rubber polymer, which is primarily regulated by rubber–bitumen interactions. Jeong et al. [26] mentioned that these interactions vary based on the bitumen composition and the rubber surface treatments, and they are sensitive to environmental variables. Additives help improve the performance of CLNR in bitumen modification. Choosing an appropriate additive varies from one area to another depending on the geographic location and resources in each country. Pavement professionals should not choose additives based solely on how well bitumen functions; instead, they should consider other aspects, such as economic concerns, the manufacture of modifiers and environmental compatibility [6]. Previous studies indicated that one modifier alone cannot enhance all the requisite pavement functional properties. Bitumen should be modified with several additives to improve diverse characteristics because of the complex interactions of modified bitumen.

### Treatment of Cup Lump 

A homogeneous mixture is crucial to ensure a uniform composition that cannot be separated physically through any separation process. Generally, the components of a homogeneous mixture retain their own properties. Polymer–bitumen compatibility is crucial to achieve a homogeneous mixture by including well-dispersed polymer into bitumen formulations. CLNR is initially solid; thus, a pre-treatment technique was suggested to promote the dispersion of CLNR into bitumen. CLNR should be pre-treated with a chemical solvent to soften it before being used as a bitumen modifier. In practice, the solvent evaporation behaviour makes it less prominent in modified bitumen, which keeps the bitumen characteristics from being significantly disrupted. Studies have examined the interaction of rubber with aromatic hydrocarbons as a chemical solvent for surface treatment, and rubber swelling to alter its shape as a sticky material [12,27]. According to previous studies, a combination of evaporation-prone chemical solvents such as toluene and xylene can be used as a rubber solvent. According to the Occupational Safety and Health Administration, toluene is a clear, colourless liquid that turns into a vapour when exposed to air at room temperature and has a distinct odour. Xylene (C_8_H_10_) is a colourless, flammable liquid with a sweet odour. Toluene and xylene may be used in bitumen. However, few studies have evaluated how their use affects bitumen modification. 

Referring to Azahar et al. [28], various rubber-to-toluene ratios at 1:1, 1:1.5, 1:2, 1:2.5 and 1:3 with 20 g of cup lump were placed in five separate steel containers with lids, to identify the optimum ratio. The samples were kept at room temperature in the lab, and CLNR’s swelling behaviour over time was observed while the rubber-to-toluene ratios and the treatment time were analysed. Steel containers with identical dimensions were employed to verify that the rate of evaporation was consistent. For the first day of treatment, the toluene loss was measured hourly to track CLNR’s physical change. The record was continued the next day until the toluene had entirely absorbed and evaporated. The 1:2 rubber-to-toluene ratio was identified as the optimum ratio at the end of the experiment. This optimum ratio corresponds to the findings of Abdul Ghafar et al. [29]. CLNR with a size of 10 mm was treated with a toluene-to-CLNR ratio of 2:1 for 24 h before being mixed with base bitumen. The same ratio was used by Mohd Azahar et al. [9]. Next, 5%, 10% and 15% rubber crumb were prepared with toluene for 24 h with a 1:2 rubber-to-toluene ratio. The blending procedure consisted of two steps. First, the treated rubber was shredded. Then, it was combined with bitumen. The treated rubber was initially placed in the centre of the supporting column, which consists of a shaft attached to the motor at one end and the head at the other. The treated rubber was rotated for 2 min at 2000 rpm with the intention of ripping the rubber surface. This method might slow down the ageing of bitumen and ensures that toluene is discharged by evaporation because of the shorter duration needed to integrate the two ingredients. Table 1 summarises the method of toluene application with commonly used percentages of rubber, comprising the tested ratio and the optimum rubber-to-toluene ratio obtained along with its soaking duration based on previous studies. 

In the application of xylene, Hazoor Ansari et al. [20] treated 50 g of dried CLNR with different CLNR/xylene ratios of 1:1, 1:2.1:3 and 1:4. All containers had equal sizes and dimensions to prevent any variation in the outcomes. All containers were stored at room temperature in a lab on a flat, level surface for 48 h. The optimum ratio was 1:3. CLNR was gradually introduced into the bitumen in increments at a shearing speed of 500 rpm to increase the xylene evaporation from CLNR. This procedure can shorten the blending time, enhance rubber absorption and eventually slow down bitumen ageing. The swelling and deterioration of the rubber often cause rubber–bitumen interactions at high temperatures. Only the broken down and disentangled CLNR chains interacted with the heated bitumen in pre-treated CLNR because the solvent already regulated the swelling. The low-molecular-weight portion of bitumen (maltenes) is disseminated and absorbed by the network of polymers when CLNR interacts with warmed bitumen at a high shearing speed and temperature; this process encourages additional xylene evaporation. Table 2 shows the method of xylene application with commonly used percentages of rubber, comprising the tested ratio and the optimum rubber-to-xylene ratio obtained along with its soaking duration based on previous studies. 

## 3. Type of Additives and Their Methods of Application

The use of additives in CLNR mainly aims to increase the homogeneity of the bitumen, thus enhancing the properties of the mixture for better pavement construction. A high CLNR content corresponds to severe mixing conditions and thus a higher viscosity of the base bitumen. This condition requires higher mixing and compaction temperatures, resulting in poor flow characteristics for the modified bitumen. Furthermore, this condition produces a large amount of polluting fumes and oxidative ageing, which will increase the overall cost of bitumen production [4,21]. Moreover, polymer has significant drawbacks, such as degradation, oxidation and susceptibility to free radicals in rubber, which accelerate bitumen ageing [32]. Therefore, for a clean manufacture of rubberised bituminous mix, the manufacturing temperature should be reduced without sacrificing mechanical performance. For this reason, a wide range of additives that can increase distinct characteristics of bitumen and work in diverse ways have been utilised in rubberised bitumen modification, such as PPA, Evotherm, MPP, TMQ and sulphur, which were identified as co-modifiers [4,12,22,23,30,33,34]. 

### 3.1. Polyphosphoric Acid

PPA is a reactive agent made of liquid mineral polymer with a series of phosphoric acid oligomers that has gained attraction due to its relatively enhanced bitumen performance at a low cost [31,35,36]. Table 3 outlines the properties of PPA based on previous studies. The addition of PPA shows good compatibility with the bitumen and can be used in conjunction with other bitumen polymer modifiers such as styrene–butadiene–styrene (SBS) and styrene–butadiene rubber (SBR), ground tire rubber (GTR), desulphurised rubber-modified asphalt (DRMA), crumb rubber (CR), and latex-modified bitumen [16,31,37,38,39,40]. Unlike SBS, economical modifiers such as PPA can improve the characteristics of bitumen. Numerous functional groups in bitumen can react with PPA [40]. It disperses the asphaltenes and makes improved interactions with rubber possible by allowing for greater dispersion of asphaltenes in the maltenes phase [41]. Using a modest quantity of PPA to create a PPA-modified bitumen will give it rheological qualities comparable with those of an SBS-modified bitumen [16]. Table 4 and Table 5 summarise the blending process and the amount of PPA used in bitumen and various types of rubber, respectively. 

Hazoor Ansari et al. [20] examined how CLNR and PPA affect bitumen properties, including its shape, rheological features, temperature susceptibility and storage stability. Nodified bitumen was obtained by adding four percentages of CLNR (3%, 6%, 9% and 12%) and PPA at 0.5% by weight of bitumen and mixing them for 30 min. After PPA was blended with base bitumen at concentrations of 0.5%, 1%, 1.5% and 2%, the optimal value was determined to be 0.5% by trial and error. After 90 min of mixing, the mixtures were held at 180 °C for an additional three hours. This method was used to ensure that there was no trapped air and to further disseminate the rubber chains throughout the bitumen for a uniform modified bitumen. The greater PPA concentration with CLNR was impractical because it also causes the base bitumen to stiffen [18]. Saowapark et al. [42] mentioned that the modified bitumen with 3.2 wt.% of NRL and 2 wt.% of PPA is a recommended formula that is suited for the road conditions in Thailand with a fixed shearing time of 30 min. In addition, Qian et al. (2019) [43] determined 1 wt.% as the ideal dosage of PPA in CR-modified bitumen at a shearing speed of 600 rpm for 30 min. This result was similar to that obtained by Yadollahi and Sabbagh Mollahosseini [41]. Bitumen was mixed with 1% PPA at 160 °C for 60 min while PPA was gradually heated to 175 °C. Many bubbles form during PPA blending, indicating that bitumen and PPA undergo a chemical reaction with some gas. Therefore, mixing should not cease until all bubbles have vanished. Then, 1% PPA was added to bitumen to catalyse the formation of low- and high-sulphur bitumen [44]. 

**Table 3 polymers-15-01951-t003:** Properties of PPA based on previous studies.

Reference	Concentration of Phosphorous Pentoxide, P_2_O_5_ (%)	Concentration of Phosphoric Acid, H_3_PO_4_ (%)
[37,45,46]	>85	-
[47,48]	-	105
[49]	-	110
[31,50]	>80	-
[13]	83.5	-
[21]	79.3	-
[51]	84	-
[22]	75.9	-
[22,52]	83.3	115
[23]	115	-

**Table 4 polymers-15-01951-t004:** Method of PPA application based on previous studies.

Reference	Type of Rubber	Blending of PPA		
		Blending Temperature (°C)	Blending Time (Minute)	Blending Speed (rpm)
[40,53]	-	150	20	4000
[22]	-	150–155	60	1200
[17]	-	135	30	245
[54,55,56,57]	SBR	160–170	20–60	4000–5000
[41]	CR	160	60	-
[51]	NRL	150	30	-
[58,59]	SBS	165–175	20–30	5000
[20]	CLNR	165	30	2000

**Table 5 polymers-15-01951-t005:** Application of PPA in rubber polymers.

Rubber Additives	Percentage Rubber	Percentage PPA	Reference
SBS	2–6%	0.25–1.6%	[39,45,58,60]
SSBR	1.5–6%	0.5–2%	[39,56]
GTR	8–16%	0.3–1.2%	[40,53]
DRMA	18–22%	1.25%	[45]
CR	5–18%	0.4–2%	[16,58]
NRL	0.6–4.5%	1–2%	[51]
CLNR	3–12%	0.5%	[20]

### 3.2. Evotherm 

Warm-mix asphalt (WMA) technology has grown in popularity in recent years because it can drastically lower the working temperature of the bitumen mixture while ensuring its performance and minimising pollutant gas emissions during the mixing and paving stages, resulting in improved on-site construction conditions [61,62,63]. Other advantages of WMA technology include lesser fuel consumption, a faster construction period and transportation time opening and less bitumen thermal ageing [30]. Evotherm warm-mix additive is a chemical package that comprises workability accelerators, adhesion promoters and emulsifiers [31]. Evotherm is a brown, oily liquid or chemical additive that is partially water soluble and has a fishy amine odour. It contains amine agents that increase bitumen workability during the mixing and compacting process. The product allows bitumen mixes to be mixed and compacted at lower temperatures without compromising quality. Table 6 outlines the properties of Evotherm based on previous studies. Table 7 and Table 8 summarise the method of blending process and the amount of Evotherm in various types of rubber, respectively.

Suleiman Abdulrahman et al. [64] conducted a study using CLNR and with 0.3%, 0.4%, 0.5%, 0.6% and 0.75% Evotherm-modified bitumen at 160 °C for 5 min to produce warm cup-lump-modified bitumen (WCMB). The optimum WCMB was identified and the ideal mixing and compaction temperatures were established through physical and rheological tests, which were conducted to evaluate the impact of this alteration by using aggregate coating and compatibility tests. A similar study was conducted by S. Abdulrahman et al. [4] but with different shearing temperature, speed and time at 170 °C and 4000 rpm for 2 h, respectively. Then, 0.5% Evotherm was selected as the optimum dosage of the warm-mix additive. Furthermore, the CR-modified bitumen in its terminal blend was studied with 0.3% dosage of Evotherm by weight of bitumen [35]. H. Yu et al. [36] used Evotherm to investigate the rheological properties of warm rubberised bitumen prepared by different mixing procedures at 160 °C for 1 h. H. Yu et al. [65] used different shearing times. Their blending process was conducted for 10 min at 160 °C with a shearing speed of 800 rpm.

### 3.3. Mangosteen Powder

Rahmah et al. [60] evaluated the effect of 2–6 g of MPP on the ageing properties of 10 g CLNR. MPP has different bioactive compounds such as phenolic acid and flavonoids, which have biological and medicinal properties. Notably, antioxidant capabilities are due to phenolic natural xanthones such as α-mangostin and γ-mangostin [45]. MPP is applied mainly because it can replace synthetic hindered phenols or amine and enhance the ageing properties of the rubber modifier [60]. MPP also helps prevent oxidative degradation [54]. Table 9 shows the size of MPP used, and Table 10 shows the amount of rubber content and the respective amount of added MPP. Limited studies correspond to the implementation of MPP in rubber polymers. Further studies are recommended to enhance the use of this renewable resource. 

### 3.4. Trimethyl-Quinoline

TMQ (2,2,4-trimethyl-1,2-H-dihydroquinoline) is an excellent antioxidant that is relatively low cost and has various uses in rubber manufacturing processes. It reacts with oxides or broken polymer chain ends generated by reaction with oxygen [47]. Its product is a combination of oligomers such as dimers, trimers and tetramers. The content of oligomers, particularly dimers, determines the quality of TMQ [48]. As a result, they prevent oxidative degradation from spreading, thus preserving the elastomer physical properties. Used polymers are usually disposed of in the environment, and can be recycled in some circumstances. Thus, additive optimisation is essential for the distinctive qualities of rubber to function and for the release of various additives into the environment to be controlled [50]. Table 11 and Table 12 display the blending process and the percentage inclusion of TMQ in CLNR, respectively. 

### 3.5. Sulphur

The application of sulphur on CLNR has not yet been reported, but its significant contribution to the enhancement of properties has been documented by previous researchers. The low cost and availability of sulphur as a by-product of the oil and gas sectors may explain the spike in interest in using it as an additive or partial alternative for bitumen [53]. Table 13 outlines the properties of sulphur based on previous studies. One of the most prominent and commonly utilised methods is to modify bitumen using chemicals such as sulphur. Since the 1940s, sulphur has been found to affect and improve the characteristics of bitumen pavements [70]. When sulphur is used as a bitumen extender, it decreases the amount of bitumen needed and greenhouse gas emissions while also changing the rheological properties of bitumen. As a result, it minimises the use of bitumen, a naturally occurring substance on the verge of depletion, and makes the mix more cost-effective because sulphur costs three to four times less than bitumen [61]. The influence of sulphur on bitumen varies depending on the bitumen’s origin, the types of sulphur used, the sulphur concentration and the blending temperature [52,71]. When the price of bitumen returned to normal, sulphur in pavement engineering became less appealing, and safety and environmental issues have been raised about processing sulphur-extended bitumen. Sulphur can accelerate the cross-linking action between the polymer and the components in the bitumen and establish a strong chemical connection, which manifests as a microscopic impact of a more uniform rubber polymer phase distribution [71]. Table 14 summarises the method of the blending process, and Table 15 shows the percentage for the inclusion of sulphur in various types of rubber content.

According to Saowapark et al. [51], 0.3 wt.% of sulphur sheared for 10 min with 3.2 wt.% of NRL was selected as the optimum concentration for the road conditions in Thailand. A similar dosage was used by S. Wang et al. (2021) [55], showing that sufficient sulphur concentration helps ensure that the network structure of the CR-modified asphalt (CRMA) is homogeneous. However, too much sulphur will prevent the desulphurisation and deterioration of the CR in the bitumen, which would harm the CRMA’s low-temperature performance. The blending process was performed under a shearing time of 120 min at 180 °C and is similar to that performed by S. Wang, Huang, Liu, Lin et al. [58]. In the application of SBS, 1% sulphur was added to 2% of SBS with stirring for 2, 4 and 8 h. Gupta et al. [56] also used sulphur as a cross-linker in the fabrication of new experimental bitumen using bitumen (70/100) and high-binyl content styrene-butadiene copolymer. Hung et al. [57] mixed rubber-modified bitumen with 3 wt.% sulphur at 135 °C for 5 min. The same shearing condition was implemented by Mousavi and Fini (2021) [53], in which 10% sulphur was blended with rubberised bitumen by weight. The same concentration of sulphur (10% by base bitumen weight) was blended at a shear speed of 3000 rpm at 180 °C for 30 min, with the bitumen produced having a high sulphur content [52,71]. Zhou et al. [59] added 10% sulphur to bio-modified rubberised (BMR) bitumen at a temperature of 155 ± 5 °C for 30 min using a mixer at 1000 rpm. Das and Panda [72] added 2% of sulphur to bitumen at various temperatures ranging from 100 to 160 °C, with a 10 °C increment for various mixing times of 15, 30, 45, 60 and 75 min.

**Table 13 polymers-15-01951-t013:** Properties of sulphur based on previous studies.

Reference	Physical State	Odour	Specific Gravity	pH Values	Boiling Point (°C)	Melting Point (°C)
[64]	Dark amber	Amine-like	1.03–1.08	10–12	>200	-
[37]	Brown liquid	-	-	-	-	-
Evotherm-DAT [65]	Caramel	Amine-like	-	9–10	150–170	-
Evotherm-3G [65]	Light- orange	Amine-like	-	8–9	150–170	-
[66]	Amber-dark	Amine-like	-	10	200	<−30
[38]	Brown liquid	-	-	9.1	-	-

**Table 14 polymers-15-01951-t014:** Sulphur application based on previous studies.

Reference	Type of Rubber	Blending of Sulphur		
		Blending Temperature (°C)	Blending Time (Minute)	Blending Speed (rpm)
[51]	NRL	150	10	-
[59]	Bio-modified rubber	155 ± 5	30	1000
[73]	-	140	30	1200
[72]	-	150 ± 5	15	3000
[44]	-	180	30	3000
[74,75]	CR	180	120	-
[62]	NR	70	10	40
[57]	Rubber-modified bitumen	135	5	-
[76,77]	SBS	190–220	30 @ 120–480	5000

**Table 15 polymers-15-01951-t015:** Application of sulphur in rubber polymers.

Rubber Additives	Percentage Rubber (%)	Percentage Sulphur (%)	Reference
NRL	3.2	0.3	[51]
CR	5–40	0.15–0.4	[74,78]
Polybutadiene Rubber/NR	70/30	1.3	[62]

## 4. Performance of CLNR with Additives

### 4.1. Xylene and Toluene

According to Azahar et al. [28], the evaporation of toluene in CLMB may be accelerated due to its tiny surface area, reducing the toluene-cup lump content. Increased solvent concentration can cause a high evaporation rate, which leads to swelling of rubber content with excess toluene on the container’s top surface, whereas low concentration results in dissatisfactory dissolution of CLNR. The optimum rubber-to-toluene ratio was 1:2 [28]. Mohd Azahar et al. [9] mentioned that CR from CLNR treated with toluene softens the bitumen, thus affecting the corresponding softening point value. The decline in penetration index (PI) was identified, which can be linked to bitumen’s poor temperature susceptibility. This finding is consistent with that of Azahar et al. [28]. The penetration value for the 1:2 rubber-to-toluene ratio decreases by about 22% and 24% at 24 and 48 h of treatment, respectively, compared with the conventional bitumen. This finding shows that a greater treatment ratio softens the rubber due to the generally high toluene concentration in the CMA. Consequently, the decrease in penetration value for the 1:2 rubber-to-toluene ratio is represented in the changing softening point, which rose by roughly 6% and 8% for 24 and 48 h of treatment time, respectively, when compared with conventional bitumen.

Azahar et al. [28] and Mohd Azahar et al. [9] found that the presence of toluene also helped reduce the viscosity value of the CMA, which led to performance enhancement without weakening its properties. A high toluene percentage affects viscosity by providing a less viscous bitumen, and adding solvent can inadvertently modify the structure of the samples. The solvent can contribute to the dissolution or aggregation of the bitumen with the heaviest or most polar molecules, leading to a less viscous bitumen depending on its type [78]. In addition, the blend of toluene with rubber and bitumen produced a stable and homogeneous mixture [9], as confirmed by Kazemi et al. [79]. They concluded that the addition of toluene strengthened the bitumen self-healing capacity and that increasing the polymer concentration intensified this enhancement [79]. When rubber additives are subjected to high temperatures during bulk storage, toluene helps rubber retain its asphaltic compounds [28]. At higher temperatures and healing time extension, the healing rate performance of modified bitumen increased. The ductility of modified bitumen decreased slightly in the presence of chemical solvents of rubber because the CMA mix was pulled as thin as a fine thread throughout the test. Hence, CMA with chemical solvent was more likely to break apart than CMA without chemical solvent [9]. Thus, toluene is one of the best chemical solvents to be used in rubber treatment. Table 16 summarises the significant properties of toluene from previous studies. 

Limited research has studied the effects of xylene in rubber modification. Hazoor Ansari et al. [20] reported that with the addition of xylene to CLMB, penetration was steadily reduced, while softening and rotational viscosity increased. In research on surface treatment and rubber swelling to change its shape as a sticky material, the interaction of aromatic hydrocarbons used as a chemical solvent with the rubber was observed [12,27]. Generally, when rubber and bitumen interact at high temperatures, the rubber swells and degrades. However, treating CLNR with xylene can minimise blending time, enhance rubber dissipation and eventually reduce bitumen ageing [20]. Table 17 summarises the significant properties of xylene from previous studies. Despite its limited application, xylene can be used to pre-treat CLNR to obtain good polymer and bitumen compatibility.

### 4.2. Polyphosphoric Acid

Few studies have focused on the utilisation of PPA with CLNR. Hazoor Ansari et al. [20] used PPA with CLNR in their studies. In general, using 0.5 wt.% PPA reduced CLNR concentration by 3% while improving bitumen performance consequently. This finding suggests that adding PPA can lower the percentage of CLNR aside from improving the compatibility and performance because of a change in the colloidal structure of the bitumen gelling agent, which increases the quantity of asphaltene and enhances bitumen stiffness [12]. PPA may also increase the homogeneity between CLNR and bitumen, showing its uniformity through dispersion analysis based on field emission scanning electron microscopy and the smooth curves obtained [20]. This finding confirmed the conclusion of the potential of mixing PPA into rubber polymers [31,60]. The inclusion of 0.5 wt.% PPA enhanced the penetration resistance and the softening point of CLNR [20]. This result confirmed previous studies, in which the incorporation of more than 1.0 wt.% PPA increased the softening point [31,35,60]. The softening point of the bitumen improved further as the PPA concentration increased, indicating a good effect of the PPA as a co-modifier that can react and link bitumen molecules, resulting in a larger aggregate size and an enhanced softening point. However, the increase in the penetration resistance of CLNR observed in the study by Hazoor Ansari et al. [20] deviated from past studies, in which the penetration value lowered as the PPA concentration increased [31,35,60]. This result occurred because of the interaction of PPA with bitumen, which produced larger-molecular-weight resins or asphaltene structures with substantially lower penetration values [36,71]. A higher PI implies better temperature susceptibility, thus proving that a small quantity of PPA can improve the temperature susceptibility and reduce CLNR content within the bitumen matrix, which is consistent with Qian et al. (2019) [43]. 

Moreover, the results reveal that combining CLNR with PPA not only improves high-temperature properties but also increases resistance to shearing at high temperatures [20]. This condition is due to the shift of the bitumen structure from sol-type to gel, at which a strong colloidal structure is generated due to the high concentration and volume of resins and asphaltenes, resulting in a change in the bitumen’s viscoelastic behaviour, hence enhancing its high-temperature performance [56,58,80]. This result agreed with previous studies that found that adding PPA to CR, SBS/SBR and DRMA can improve the high-temperature performance of the composite bitumen [16,31,59,72]. PPA also helped to enhance the stiffness of CLNR through the process of deagglomeration and conversion of resins into asphaltenes with even dispersion within the maltene matrix [20]. This finding is consistent with those of Saowapark et al. [51] and Qian et al. [43], who found that rubber was made stiffer by adding a PPA modifier mainly due to the structural shift of bitumen from sol to gel. The viscosity of the modified bitumen increased, confirming that PPA is a reactive additive and can be used to enhance the viscosity of a composite [22,31,60].

Furthermore, PPA successfully improved the elastic response of CLNR according to the obtained isochronal plots [20]. Elastic response, also known as elastic recovery, refers to a composite’s ability to return to its original state once the load has been removed. This outcome corresponds to the studies of Qian et al. [43], Wistuba et al. [49] and Ma et al. [23]. The inclusion of PPA resulted in a lower difference in softening point between the top and bottom of test tubes due to the interaction between PPA and bitumen, which increases viscosity and molecular weight and hence delays the phase separation of CLNR and bitumen [22,31,60]. The phase separation during the thermal storage associated with the PPA content and composition of base bitumen show improved storage stability, confirming the findings of Liu et al. [14] and C. Li et al. [80]. The Fourier-transform infrared spectroscopy (FTIR) spectrum of the PPA addition to CLNR revealed additional peaks, indicating that the PPA and bitumen undergo a chemical reaction with the –OH hydroxyl group to generate phosphate ester, implying that the modification is primarily a chemical and physical process [22,31,60].

Table 18 summarises the significant properties of PPA from previous studies. Previous studies that incorporated PPA with other rubber polymers (NR, CR, NRL, DRMA, SBR, etc.) showed that the material exhibits improved toughness and tenacity, enhanced viscoelasticity, rutting resistance, resistance to permanent deformation, fatigue resistance with reduction in creep stiffness, and ductility [16,31,55,59,71,81]. PPA also helped increase the high- and low-temperature performance of modified bitumen [31,35,38,81], endowed it with good anti-ageing behaviour [72,73,81] and improved the adhesion force of base bitumen [15]. Thus, PPA can be considered a good alternative to polymers.

### 4.3. Evotherm 

The utilisation of Evotherm in CLNR is receiving attention due to its various beneficial features. S. Abdulrahman et al. [4] reported that adding 0.3–0.4% Evotherm softens CLMB by increasing its penetration. This outcome corresponds to the studies of H. Yu et al. [36] and X. Yu et al. [37]. The introduction of Evotherm resulted in a better penetration value because of the nature of Evotherm liquid, and it had a softening effect on the bitumen, making it more fluid and increasing penetration [82]. The softening point was also improved by adding Evotherm [4,83]. Aside from improving the compatibility, Evotherm also improved the workability of the bitumen mixture with smaller air void contents [23], [79]. Moreover, switching CMA from hot-mix asphalt to WMA lowered the mixture’s fuel consumption and carbon emissions considerably, reducing its global warming potential by 5 kgCO_2_e [64]. The environmental advantages of using Evotherm WMA include reduced consumption of neat bitumen with the inclusion of CR, fuel savings and lower hazard emission [35]. Evotherm also lowered the mixing and compaction temperature, thus decreasing CLNR’s viscosity [4]. This condition occurred because the warm-mix additive mainly serves as a surfactant, lowering the surface tension between the aggregate and the bitumen. This finding is consistent with that of H. Yu et al. [36].

In addition, higher tensile strength, best anti-stripping performance and better moisture damage resistance were reported when using Evotherm [35]. Evotherm contains anti-stripping agents that improved the interface of the mixture. The effect of the warm-mix agent resulted in lesser rutting depth as the bitumen mixture became easier to compact even at temperatures that are considerably lower than the field compaction temperature [79]. Evotherm helps improve the fatigue lives of the composite [42]. In general, Evotherm has better fatigue crack resistance due to a better aggregate interface, consequently reducing the hardening degree of warm-mix high-viscosity asphalt (HVA) during the long-term ageing process, thus allowing the energy to be dissipated by rebound deformation without cracking [83]. HVA with the chemical WMA additive (Evotherm) has a lower oxidation sensitivity throughout the ageing process [51,69,78]. According to the disc-shaped compact tension test, the use of warm-mix additive provides better low-temperature performance in terms of fracture energy and peak load [35]. 

Evotherm, on the other hand, improves the interaction with bitumen, making it stiffer and less vulnerable to shear stress [36]. Any sort of WMA additive can considerably reduce the rotational viscosity, toughness and tenacity of warm-mix HVA because of decreased surface free energy and the capacity of Evotherm to reduce the flexibility and anti-deformation ability of HVA [37]. Evotherm also increases the absolute viscosity and poor performance at high temperatures. When rubber elastomers interact with Evotherm, a chemical cross-linking structure forms in the bitumen, preventing polymer molecules from moving freely, resulting in an increase in absolute viscosity [37]. Nevertheless, Evotherm may cause a reduction in the mixture modulus and poor resistance to permanent deformation [65], aside from reducing ductility [37]. Table 19 summarises the significant properties of Evotherm from previous studies. Findings indicate that Evotherm plays a great role in enhancing the performance of rubber elastomers.

### 4.4. Mangosteen Powder

Adding MPP enhances the compound ageing properties [84]. This finding confirms previous studies that found that rubber ageing properties and oxidation resistance were successfully enhanced by adding MPP rich in polyphenols as a UV interceptor [30,85]. MPP is homogeneous and suitable to be added to NRL compounds [86]. Adding MPP provides good mechanical properties to rubber, such as high percentage of elongation at break, high tensile strength, good stiffness, high Young’s modulus, high wettability, dense film without water leakage and good contact angle [74,78,87,88] MPP also protects against free radical attack and degradation. A high level of MPP can serve as an antioxidant, and FTIR showed that its performance as an excellent sealant was not likely impaired by ageing due to weather or thermal exposure [60]. MPP is considerably more beneficial than heat ageing at promoting weight retention, especially for weather exposure. Table 20 summarises the significant properties of MPP from previous studies. Despite the minimal studies by previous researchers, MPP has great potential for application as an antioxidant improving the properties of rubber polymers.

### 4.5. Trimethyl-Quinoline 

Ibrahim et al. [32] studied solid natural crepe rubber from cup lump ranging from 8% to 12 wt.% with the addition of antioxidant TMQ (1%, 2% and 3% of the total sample weight) to produce bitumen with good durability and high resistance to rutting and low or moderate temperatures and to avoid premature ageing of bitumen. The increased ratio of crepe rubber to TMQ in the modified bitumen tends to promote penetration, resulting in a lower softening point. Furthermore, the fluid character of the TMQ melt, which possesses aromatic and aliphatic molecular structures comparable to maltene compounds in bitumen, softens the consistency of the mixture [90]. Increasing the TMQ ratio results in the limited interaction of rubber in the bitumen, resulting in softer modified bitumen [91]. TMQ functions to prevent modified bitumen oxidation; thus, adding it to the mix helps improve the bitumen–aggregate bond [32]. Nonetheless, due to its physical properties, such as waxiness, the addition of TMQ resulted in decreased viscosity and rutting factor, as well as fluctuation of penetration value and increasing softening point, because of the depletion of cohesion and adhesive capabilities of bitumen. This finding is associated with the rise in Marshall stability, indicating that the sample is rutting- and fatigue-resistant [32].

Furthermore, an enhancement in the mechanical properties of rubber composite was reported with the addition of TMQ [33,74]. TMQ plays an important role in preventing rubber or bitumen oxidation over its life span and storage either by stabilising or scavenging free radicals or by suppressing peroxide formation [92]. The decrease in phase angle resulted in a rise in elastic proportion in the bitumen-rubber combination, implying that TMQ plays an important role in the viscoelasticity of the bitumen [93]. Table 21 summarises the significant properties of TMQ from previous studies. Overall, TMQ improves the longevity of bitumen and works well in strengthening the resistance of bitumen to ageing in both short- and long-term ageing conditions [32].

### 4.6. Sulphur

Previous studies have reported that as low as 0.3% sulphur can partially congeal into crystalline sulphur particles, thus improving toughness and tenacity with additional strength [61]. Sulphur particle dispersion results in the formation of additional polysulphuric networks in the bitumen phase, which establish the cross-linking structure and contribute to improved storage stability with no phase separation at high temperatures and acceptable rheological analysis results [71,73]. Furthermore, scanning electron microscopy images revealed that small sulphur particles were appropriately dispersed into the base bitumen, homogenising the bitumen and demonstrating the compatibility of the modifier with bitumen [74]. As a result of the intermolecular interaction between the bitumen and sulphur, the bitumen characteristics improved [73]. Sulphur vulcanisation reveals that NR and bitumen are compatible, which helps minimise the probability of NR droplets coalescing during mixing and supports effective dispersion of the polymer-rich phase in modified bitumen. As time passes, sulphur continues to react with bitumen components, forming an internal network of organic molecules, while excess sulphur crystalises into micrometre-sized structures, resulting in improved stiffness parameters, particularly in the low-frequency range, and softening the bitumen mixture [61]. 

With increased sulphur solubility and less free sulphur, a more homogeneous cross-linked network structure was observed, thereby increasing the interactions between the filler and the matrix, which is highly dependent on bitumen composition [59]. This condition improves thermal conductivity and reduces heat build-up performance, while also enhancing tensile and tear properties. Enhanced cross-link density and a more uniform rubber network that can withstand greater force and resist breakup resulted in a thicker and more uniform rubber wall that strengthens the cross-linking effect between the polymer and the bitumen components and forms a stable chemical bond, promoting polymer homogeneity [71]. Higher temperature ageing resistance and the capacity of sulphur to prevent polymer molecules from segregating in bitumen strengthened the stability and anti-ageing property of bitumen, resulting in improved short- and long-term ageing resistance [71,80]. When sulphur is incorporated into the cross-linked network structure of the polymer, the hardening rate of the bitumen during the ageing stage is slow [71].

Sulphur inclusion is attributed to the conditions of the internal molecular structure. Sulphur rearrangement in the bitumen matrix results in increased resistance to oxidative ageing, surface roughness, fatigue resistance and viscosity of sulphur-modified bitumen [71]. Moreover, adding sulphur to the base bitumen increased its high-temperature performance [61]. Singh et al. [61] stated that modified bitumen has a considerable decrease in ductility and a greater Marshall stability than conventional bitumen. Importantly, sulphur elevated the properties of all hybrid polymer-modified bitumen, and FTIR data demonstrated absorption peaks, confirming the efficiency of sulphur as a cross-linking agent for enhancing chemical interactions between polymers and bitumen molecules. The continuous network structure that promotes the adhesion between the mixture that continues to cure and bond more with bitumen, thus acquiring elasticity, exhibited improved elastic behaviour especially at higher temperatures [68,71,73]. Additional chemical reactions and C-S-C bonds that developed during the thermal ageing phase reinforced the rheological behaviour, with improved resistance to rutting and increased thermal stability.

However, previous studies found a wide range of softening points and penetration values. According to Saowapark et al. [51], the improvement in softening point was due to NR/PPA-modified bitumen with increased sulphur levels being more susceptible to thermally oxidative ageing, causing bitumen degradation under the influence of heat [61]. Singh et al. [61] and Iqbal et al. [73] recorded a reduction in the softening point along with its penetration value due to stiffness caused by sulphur pallets. The inclusion of 36% sulphur in the bitumen has a plasticising impact on the pavement, indicating higher resistance to thermal cracking at low temperatures and lesser permanent (plastic) deformation at high temperatures [73]. Mousavi and Fini [53] reported that adding sulphur reduced the stiffness of bio-modified rubberised bitumen, thus negatively affecting bitumen performance. Table 22 summarises the significant properties of sulphur from previous studies. Hence, the addition of sulphur helps strengthen the bitumen’s properties, especially in producing WMA pavement. 

## 5. Conclusions

NR is one of the renowned renewable resources that have high potential for use in bitumen modification. The addition of additives or modifiers is significant for further enhancing the performance of rubberised bitumen. This research highlights previous findings on the use and consequences of potential additives (PPA, Evotherm, xylene, toluene, MPP, TMQ and sulphur) in bitumen modification with NR, with the goal of extending the service life of roads while lowering maintenance costs and increasing ride quality. 

In rubber treatment, xylene and toluene with rubber-to-solvent ratios of 1:3 and 1:2, respectively, effectively produced a homogeneous rubber mixture to be blended with bitumen. Xylene is less commonly used than toluene. Further studies are needed to evaluate the efficiency of xylene as a solvent in rubber treatment. PPA is an important bitumen modifier and can be used with rubber types such as SBR, SBS, CR, NRL and CLNR to improve the physical and mechanical properties of the mixture. The amount of 0.5% PPA is effective for CLNR, which is the main focus of this study. Meanwhile, 1.2–1.6% and 2% of PPA are compatible with CR and NRL, respectively [20,23,51]. In addition, the use of Evotherm in rubber-modified bitumen is common in WMA technology with a lower working temperature. The implementation of WMA is beneficial for reducing emissions. Thus, its adoption by the road industry would lead to sustainable construction. Moreover, 0.5% of Evotherm improved the performance of CLNR-modified bitumen [4]. New additives are being studied with CLNR, such as MPP and TMQ. Further studies are needed to verify the application of these additives to enhance the properties of modified bitumen. Higher content of MPP (6 g) and 2% of TMQ was effectively used with CLNR to enhance the properties of CLNR-modified bitumen [32,89]. Currently, no study has implemented the use of sulphur with CLNR. However, sulphur makes significant contributions to other types of rubber, such as NRL, CR and SBS, because it helps reduce the amount of bitumen needed, thus leading to a more cost-effective solution. Adding 0.3% sulphur to NRL and CR helps improve the properties of rubberised bitumen [32,55]. 

On the basis of this review, recommendations and considerations for further research on the application of additives with NR are listed below. 

The physical and chemical interaction between different types of NR and each additive should be investigated further;Further studies on the implementation of CLNR as a bitumen modifier should be conducted with the addition of additives because studies on the use of CLNR in bitumen modification are limited;The method of blending (blending speed, time, temperature) for each additive should be investigated further, and the optimum value should be identified for adoption in current engineering practice;The optimum ratio or dosage of each additive in the utilisation of NR should be determined to serve as a guideline for future applications.Further research on the use of CLNR with the optimum blending method and dosage should be conducted to evaluate its physical, chemical and rheological properties together with its field testing.

## Figures and Tables

**Figure 1 polymers-15-01951-f001:**
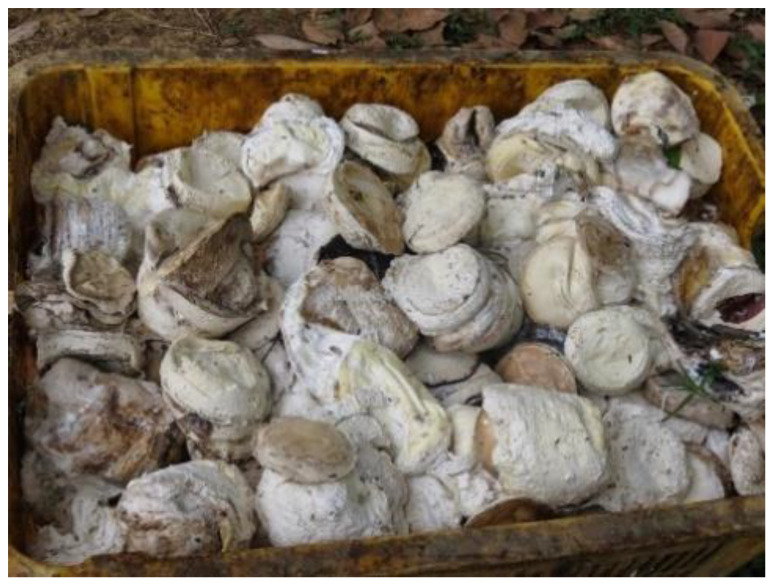
Cup lump rubber.

**Table 1 polymers-15-01951-t001:** Toluene application method based on previous studies.

Type of Rubber	Percentage Rubber (%)	Rubber to Solvent Ratio	Optimum Ratio	Soaking Duration (Hour)	Reference
CLNR	2.5–15	1:1, 1:1.5, 1:2, 1:2.5 and 1:3	1:2	24–48	[4,12,23,30,31]

**Table 2 polymers-15-01951-t002:** Xylene application method based on previous studies.

Type of Rubber	Percentage Rubber (%)	Rubber-to-Solvent Ratio	Optimum Ratio	Soaking Duration (Hour)	Reference
CLNR	3–12	1:1, 1:2, 1:3 and 1:4	1:3	48	[20]

**Table 6 polymers-15-01951-t006:** Properties of Evotherm based on previous studies.

Reference	Physical State	Odour	Specific Gravity	pH Values	Boiling Point (°C)	Melting Point (°C)	Density (g/cm^3^)	Water Solubility
[64]	Dark amber	Amine-like	1.03–1.08	10–12	>200	-	-	-
[37]	Brown liquid	-	-	-	-	-	1.2	-
Evotherm-DAT [65]	Caramel	Amine-like	-	9–10	150–170	-	>1.0	Partially soluble
Evotherm-3G [65]	Light- orange	Amine-like	-	8–9	150–170	-	>1.0	Partially soluble
[66]	Amber-dark	Amine-like	-	10	200	<−30	0.99	Partially soluble
[38]	Brown liquid	-	-	9.1	-	-	1.012	-

**Table 7 polymers-15-01951-t007:** Method of Evotherm application based on previous studies.

Reference	Type of Rubber	Blending of Evotherm		
		Blending Temperature (°C)	Blending Time (Minute)	Blending Speed (rpm)
[66,67,68]	CR	140–170	10–60	120–500 (high shear mixing)
[4]	CLNR	160	5	–
[65]	Asphalt rubber	160	10	800
[66]	GTR and SBS	150	5	–

**Table 8 polymers-15-01951-t008:** Application of Evotherm in rubber polymers.

Rubber Additives	Percentage Rubber (%)	Percentage Evotherm (%)	Reference
CLNR	2.5–10	0.3–0.75	[4,21]
CR	5–15	0.1–0.7	[66,69,70]
Asphalt rubber	-	0.5–5	[65]
GTR	8	0.5	[66]
SBS	4	0.5	[66]

**Table 9 polymers-15-01951-t009:** Application of MPP based on previous studies.

Reference	Type of Rubber	MPP Sieve Size
[60]	CLNR	50 microns
[46]	NR	75 microns

**Table 10 polymers-15-01951-t010:** Application of MPP in rubber polymers.

Rubber Additives	Rubber Amount	MPP Amount	Reference
CLNR	10 g	2–6 g	[60]
NR	-	5–20 pphr	[46]

pphr = part per hundred rubber.

**Table 11 polymers-15-01951-t011:** Method of TMQ application based on previous studies.

Reference	Type of Rubber	Blending of TMQ		
		Blending Temperature (°C)	Blending Time (Minute)	Blending Speed (rpm)
[32]	CLNR	165	30	300

**Table 12 polymers-15-01951-t012:** Application of TMQ in rubber polymers.

Rubber Additives	Percentage Rubber	Percentage TMQ	Reference
CLNR	8–12%	1–3%	[32]

**Table 16 polymers-15-01951-t016:** Significant properties of toluene inclusion.

Properties	Explanation	Reference
Softening and penetration	Softened asphalt binder, thus increasing the softening point by 6–8% and reducing the penetration value by 20–24%	[9,28]
PI	24% reduction of PI correlated with low temperature susceptibility with increasing treated rubber content	[9]
Viscosity	Reduced viscosity value with increasing temperature at 165 °C by 63%	[9,28]
Ductility	Reduced ductility by 29–80%	[9]
Homogeneity	Storage stability passed the specification (<2.5 °C) showing stable and homogeneous bitumen mixture	[9,66]

**Table 17 polymers-15-01951-t017:** Significant properties of xylene inclusion.

Properties	Explanation	Reference
Penetration and softening point	Increased penetration resistance (14–30%) with increasing softening point (12–25%)	[20]
Viscosity	Increased viscosity at 135 °C (72–440%) and reduced viscosity at 165 °C (5–62%)	
Homogeneity	Homogeneous modified bitumen with uniform cross-linking	

**Table 18 polymers-15-01951-t018:** Significant properties of PPA inclusion.

Properties	Explanation	Reference
Softening point and penetration value	Increased softening point by 8–24% and improved penetration resistance by 6–20%, further increasing the PI, thus implying better temperature susceptibility.	[20,41,42,51]
High-temperature performance	Enhanced high-temperature performance related to the transition of the structure of the bitumen from sol to gel, reflecting higher rutting resistance; improved the G* (13–20%) of modified asphalt; reduced phase angle δ; increased softening point (8–24%) and reduce penetration value (6–20%).	[14,20,43,49,51]
Viscosity	Increased viscosity (15–19%) of the modified bitumen.	[42,51]
Stiffness	Stiffness parameter of modified bitumen higher/better than that of the neat bitumen because of deagglomeration of the asphaltene and even dispersion within the maltene matrix; more visible at lower frequency.	[20,42,51]
Elasticity	Enhanced elastic response at lower temperature related to increasing stiffness and reduction of phase angle (21%).	[20,23,49,51]
Homogeneity	Improvement in homogeneity between CLNR and bitumen shows good compatibility/reduction of phase angle value (62–71°); CLR embedded in bitumen as shown in FESEM and EDX analyses.	[20,42,51]
Storage stability	Improved storage stability (difference in softening value ≤ 2.5 °C).	[14,20,42,51,82]
Rutting resistance	Resistance to rutting was dramatically improved; G*/sinδ increased with failure temperature 76–82 °C from 70 °C.	[51,83]
Permanent deformation	Improved resistance to permanent deformation owing to increase stiffness and elasticity; lower phase angle (70°).	[23,51]
Fatigue performance	Improved fatigue performance with decreased value of G.sinδ and fatigue failure temperature 11–13.2 °C.	[14,51]
Temperature susceptibility	Enhanced low- and high-temperature performance.	[14,41,51,68]
Ageing behaviour	Good anti-ageing behaviour and increased retained penetration ratio (≥57%) but decreased softening point increment and mass loss rate.	[23,68,82]

**Table 19 polymers-15-01951-t019:** Significant properties of Evotherm inclusion.

Properties	Explanation	Reference
Workability	Improved workability of bitumen modified with reduced air void contents (4%) owing to increasing resilient modulus (16%).	[64,84]
Environmental benefits	Reduced fuel consumption and carbon emissions, thus cutting down the global warming potential by 5 kgCO_2_e.	[35,84]
Penetration values and softening point	Increased penetration (16–22%) and softening point values (27–31 °C).	[4,36,37]
Viscosity	Decreased viscosity by 10–14% at 135 °C led to lower mixing and compaction temperature.	[4,36,37]
Rutting	Lower rutting depth than HMA (44–51%) with softer binder.	[35,36,37]
Fatigue	Improved fatigue property (2352–2521 cycles) because of better asphalt–aggregate interface.	[35,37,78]
Temperature susceptibility	Better high- and low-temperature performance.	[35,37,38]
Tensile strength	Higher tensile strength (35%).	[35]
Moisture	Better moisture damage resistance because of anti-stripping agents in Evotherm; higher TSR values (35%).	[35]

**Table 20 polymers-15-01951-t020:** Significant properties of MPP inclusion.

Properties	Explanation	Reference
Ageing	Improvement in ageing property with effective resistance to oxidation	[46,47,89]
Mechanical properties	Reduction of tensile strength (32–53%), elongation at break (20–30%), swelling percentage (≤11%) and increasing modulus (≤10%) at 100% elongation	[46,74,85,86,89]
Weight retention	Improved weight retention (tensile strength 38–80%, elongation at break (27–52%) and modulus at 100% elongation (1–5%)	[60]
Safety	Non-toxic material	[89]

**Table 21 polymers-15-01951-t021:** Significant properties of TMQ inclusion.

Properties	Explanation	Reference
Mechanical and physical properties	Enhanced mechanical and physical properties.	[32,94]
Durability	Improved durability of modified bitumen (Marshall stability 1403.96 kg with optimum asphalt content of 5.50%).	[32]
Ageing	Increased resistance to long-term ageing due to absence of carbonyl group after addition of TMQ.	[32]
Penetration and softening point	Increased (2–11%) and decreased (1–12%) penetration value with increasing (9–22%) softening point.	[32]
Rutting	Reduced rutting factor (from 6.91 kPa to 16.1 kPa).	[32]

**Table 22 polymers-15-01951-t022:** Significant properties of sulphur inclusion.

Properties	Explanation	Reference
Toughness, tenacity, strength	Improvement in toughness (12–18%), tenacity (25%) and additional strength.	[42,59]
High-temperature performance	Improvement in high-temperature performance.	[42,71,81]
Storage stability	Enhancement in storage stability (difference in softening value ≤ 2.5 °C).	[42,55,60,71]
Stiffness	Better or improved stiffness parameter resulted in improved elastic behaviour of asphalt.	[42,59,71,72]
Softening and penetration	Increase softening (12–24%) and decrease penetration (19–35%).	[42,60,72]
Ageing	More resistant to oxidative, short- and long-term ageing.	[72,80,81]
Homogeneity	Homogeneous mixture with great compatibility of modifier with bitumen; more uniform cross-linked network structure.	[42,44,55,71,72,77,81,95,96]
Elasticity	Enhanced elastic behaviour due to sulphur cross-linking effect; decreased Jnr (26–30%) and δ and increased R and G*.	[55,59,80,95]
Thermal stability and conductivity	Enhanced thermal stability and conductivity.	[61,95]
Rutting	Resistant to rutting; improved G*/sinδ (40–45%) with increasing failure temperature (2.5–4.6 °C).	[71,81,95]
Ageing	Decrease aging index (12–16%).	[55,72]
Fatigue	Enhanced fatigue resistance; reduced G*sinδ values (25%).	[59]

## Data Availability

Not applicable.

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
