# Peer review of "Potential Additives in Natural Rubber-Modified Bitumen: A Review"

_polymers, 2023, doi:10.3390/polym15081951_

Round 1

Reviewer 1 Report

The author has given reasonable answers to the questions raised, and I agree with the publication of this paper

Author Response

Thank you very much for your time reviewing my paper. 

Reviewer 2 Report

Polymers

Potential additives in natural rubber modified bitumen: a review

Comments:

* The abstract should be rewritten with main review objectives and conclusions or summaries.

* In some tables, the main properties of complex modified asphalt binders can be listed for better comparison.

* Some tables are not necessary, for example Table 15, Table 17.

* More references regarding the modification mechanism of natural rubber in bitumen should be included in this review work. For example: Investigating the role of swelling-degradation degree of crumb rubber on CR/SBS modified porous asphalt binder and mixture. Construction and Building Materials. 2021, 300, 124048.

* The English editing of the whole manuscript has to be enhanced carefully.

* The conclusion should be more concise, and more significant recommendations for future work should be supplemented.

Reviewer 3 Report

The artile presents very interesting literature review on the application and performance of using Natural Rubber in the asphalt mixtures.

The amount of work done is very big, but still there should be done some improvements for better usability. 

I have two main remarks which are applied to the whole manuscript:

1. Why dividing the information regarding amounts of materials/modifiers added on two separate tables and chapters in the manuscript? It applies to the all modifiers. It rather should be in the first table, or the table from further chapter should be moved to the "production" chapter

2. In table 12 and further, more detailed "explanations" would be gratefuly appreciated. It is now presented in the text of the manuscript, but good review articles should all its data in the tables - it graetelly improve its usability - because statements such as "Good resistance to high temperature" does not give proper information.Is good 2% improvement or 200%. Instead for example for "Increased rotational viscosity" it should rather by "Increased rotational viscosity by .....%". General descriptions as presented in the article are of no use for audience.

Also there is small editorial remark: please check tables for formating of "oC" designation

Round 2

Reviewer 2 Report

It can be accepted.

Author Response

Thank you very much for your time. 

Reviewer 3 Report

Most of the remarks were corrected, and now the manuscript is much better, but one with more detailed information on properties was done only in selected properties and tables. Most of them are still described in general way. Please consider if to extend the corrections also for other tables as stated in the first review.

Author Response

Revised accordingly. The tables (Table 16-22 (highlighted)) have been updated following the comments given by reviewer with details information of each properties mentioned. 
